# Modeling Intestinal Stem Cell Function with Organoids

**DOI:** 10.3390/ijms222010912

**Published:** 2021-10-09

**Authors:** Toshio Takahashi, Kazuto Fujishima, Mineko Kengaku

**Affiliations:** 1Suntory Foundation for Life Sciences, Bioorganic Research Institute, Kyoto 619-0284, Japan; 2Institute for Integrated Cell-Material Sciences, Institute for Advanced Study, Kyoto University, Kyoto 606-8302, Japan; kazuto.fujishima.b73@gmail.com (K.F.); kengaku@icems.kyoto-u.ac.jp (M.K.); 3Graduate School of Biostudies, Kyoto University, Kyoto 606-8501, Japan

**Keywords:** organoid, crypt, intestinal epithelial cell (IEC), intestinal stem cell (ISC), acetylcholine (ACh), cholinergic system

## Abstract

Intestinal epithelial cells (IECs) are crucial for the digestive process and nutrient absorption. The intestinal epithelium is composed of the different cell types of the small intestine (mainly, enterocytes, goblet cells, Paneth cells, enteroendocrine cells, and tuft cells). The small intestine is characterized by the presence of crypt-villus units that are in a state of homeostatic cell turnover. Organoid technology enables an efficient expansion of intestinal epithelial tissue in vitro. Thus, organoids hold great promise for use in medical research and in the development of new treatments. At present, the cholinergic system involved in IECs and intestinal stem cells (ISCs) are attracting a great deal of attention. Thus, understanding the biological processes triggered by epithelial cholinergic activation by acetylcholine (ACh), which is produced and released from neuronal and/or non-neuronal tissue, is of key importance. Cholinergic signaling via ACh receptors plays a pivotal role in IEC growth and differentiation. Here, we discuss current views on neuronal innervation and non-neuronal control of the small intestinal crypts and their impact on ISC proliferation, differentiation, and maintenance. Since technology using intestinal organoid culture systems is advancing, we also outline an organoid-based organ replacement approach for intestinal diseases.

## 1. Introduction

The small intestine is composed of crypt and villus structures (Figure 1). Villi are specialized for digestion and absorption and crypts project on the outside of villi. The two structurally and functionally different parts of the intestine are composed of a layer of IECs and lamina propria, and the small intestinal epithelium is largely renewed every 3 to 5 days [1]. Villi continuously shed differentiated cells from their tips, and these losses are replenished by intestinal stem cells (ISCs) located in the crypt bottom.

The crypt bottom harbors slender, cycling crypt base columnar (CBC) cells, which were historically proposed to be ISCs [2]. Exploiting the expression of the Wingless/Int (Wnt) target gene *leucine-rich repeat containing G protein-coupled receptor 5* (*Lgr5*) in CBCs, and genetic labeling of *Lgr5*-positive (*Lgr*5^+^) crypt cells indeed demonstrated that these long-lived cells give rise to five terminally differentiated IECs (absorptive enterocytes, mucus secreting goblet cells, Wnt secreting Paneth cells, hormone secreting enteroendocrine cells, and immune and neuronal tuft cells) via transit-amplifying cells (Figure 1) [3]. Besides actively cycling ISCs, “ + 4” position ISCs located at position +4 from the crypt bottom are a potentially distinct population of slow-cycling ISCs that are marked by *B-cell-specific Moloney murine Leukemia virus integration site 1* (*Bmi1*) (Figure 1) [4], *homeodomain-only protein homeobox* (*Hopx*) [5], *leucine rich repeats and immunoglobulin like domains 1* (*Lrig1*) [6], and *telomerase reverse transcriptase* (*Tert*) [7]. Potten and coworkers reported the existence of label-retaining cells residing specifically at this position for the first time [8]. The +4 position ISCs are resistant against acute and genotoxic stress, and have the ability to replace damaged CBC cells in response to injury for tissue regeneration [7,9].

Instead of constituting irrevocably separated lineages, it has been suggested that *Lgr*5^+^ and +4 position ISCs (*Bmi1*^+^ ISCs) can interconvert (Figure 2). *Lgr*5^+^ ISCs exhibit exquisite sensitivity to canonical Wnt signaling, contribute robustly to homeostatic regeneration, and are quantitatively ablated by irradiation [10]. Thus, *Lgr5*^+^ ISCs appear to be the hard workers of daily intestinal renewal [10]. In contrast, +4 position ISCs are insensitive to Wnt signaling, contribute weakly to homeostatic regeneration, and are resistant to high-dose radiation injury [10]. Therefore, slowly cycling “reserve” +4 position ISCs can be recalled to *Lgr5*^+^ ISC status and vice versa [5,11]. Hence, IEC system is mainly maintained by the two type of stem cells with long- and short- term regenerative potential, and the balance of the two ISC function is important for maintaining intestinal homeostasis.

Three-dimensional (3D) intestinal organoids are organized by crypt cells from mouse small intestines with specific conditions (Figure 3). Intestinal organoids are embedded in Matrigel and cultured with biochemical factors such as epidermal growth factor (EGF), Noggin (a bone morphogenetic protein inhibitor), and R-spondin 1 for mimicking the in vivo ISC niche [12]. Curiously, a single ISC including an *Lgr5*^+^ or *Bmi1*^+^ stem cell can also generate a crypt-villus domain containing spherical bodies in the absence of an intestinal epithelial and mesenchymal niche [10,13]. The *Lgr5*^+^ and/or *Bmi1*^+^ stem cells can grow in a 3D Matrigel that contains the secretions of a sarcoma cell line and many of the basement membrane proteins. Then, combined stimulation with EGF, Noggin, and R-spondin 1 leads to the formation of self-organizing bubble-like epithelial structures containing crypt-villus domains, stem cells in the bottom of the crypt, Paneth cells, enteroendocrine cells, enterocytes, goblet cells, and tuft cells. Villi protrude into the inner lumen of the organoids that are filled with apoptotic cells shedding from the villus tips (Figure 3A–F and Appendix A). The evagination of the spots to form crypts is coupled with ISC migration and proliferation and Paneth cell differentiation (Figure 3G,H and Appendix A) [14]. The evidence for these steps has also been gained by examining the formation of crypts in developing intestines or during intestinal tissue regeneration, and during continuous displacement of the intestinal epithelium [1,15,16].

The autonomic nervous system in the intestine is comprised of the sympathetic nervous system, the parasympathetic nervous system, and the enteric nervous system (ENS), which is the intrinsic nervous system of the gastrointestinal tract. Various neurotransmitters are released from ENS neurons. Since one of the major pathways of excitatory transmission within the ENS is mediated by cholinergic transmission [17], neuronal acetylcholine (ACh) potentially has functional effects on crypt homeostasis. ACh transmits a signal to recipient cells, and the key receptors that transduce the ACh message are the muscarinic and nicotinic ACh receptors (mAChRs and nAChRs). The cholinergic receptor system has been associated with intestinal epithelial cell proliferation as detailed in earlier reports [18,19]. In addition to neuronal ACh, non-neuronal ACh produced by intestinal epithelium has been found to play a critical role in maintaining homeostasis and inhibiting the differentiation of *Lgr5*^+^ ISCs via mAChRs [20]. Moreover, using an organoid system, it has been proven that ACh activates the α2/β4 nAChR subtype, which is localized with Paneth cells and modulates the expression of non-canonical Wnt ligands, including Wnt5a and Wnt9b, followed by enhancement of proliferation and differentiation in the stem cell niche [21].

Researchers have made remarkable progress in the field of organoid biology during the past decade, and organoid technology has already been reviewed in detail elsewhere [22,23,24,25,26,27]. Here, we critically appraise the advantages of organoids as model systems for research into the neuronal and non-neuronal cholinergic system in mammalian intestines. We also highlight the potential applications of organoids to meet the challenge of heterotopic transplantation, which is the transplantation of small intestinal organoids into the colon. Success will allow for adult small intestinal stem cells to be a source for cell therapy of intestinal diseases.

## 2. ISC Function Controlled by the Enteric Nervous System

The ENS is a network of neuronal, glial, and progenitor cells embedded inside the wall of the gastrointestinal tract that is responsible for the regulation of all sensory and motor function of the intestine (Figure 1) [28]. The ENS can function independently of the brain. Thus, it is frequently termed “the brain in the gut”. The ENS is considered a part of the ISC niche [29], and mucosal afferent nerves control ISCs and progenitor cells [30,31]. Therefore, understanding the role of the ENS in the intestinal crypts is of key importance.

### 2.1. Organoid-Derived Two-Dimensional Epithelial Monolayers

To parse the specific roles of ENS and intestinal epithelial communication in vivo and in vitro is of important. An in vitro optimal culture method that closely reproduces the in vivo ENS and intestinal epithelial cell composition and function has been explored [32,33,34,35]. Caco-2 cells, a human colon cancer-derived cell line, have been broadly used and have made a major contribution to pharmacological studies. However, there are problems in that some drug-metabolizing enzymes and drug transporters are absent or poorly expressed in Caco-2 cells [36,37].

Organoids have a three-dimensional (3D) morphology that is similar in arrangement to most major epithelial cell types, and can be cultured with or without an ENS component. However, as they form an inaccessible luminal compartment, it is difficult to simulate orally-administrated drugs. To resolve this problem, cell models that can precisely recapitulate the in vivo human and mouse intestinal monolayer have been investigated. Wang and coworkers generated monolayer models by using human colonic or rectal intestinal organoids, and they applied the model to study the impacts of dietary compounds on epithelial proliferation [38]. As ISC-derived organoids recreate in vivo physiology of the original tissue, this system is valuable to model medical disorders such as infectious disorders. In and coworkers showed that human colonoid monolayers are a suitable model to study Shiga toxin-producing enterohemorrhagic *Escherichia coli* (EHEC) intestinal colonization and to characterize the molecular mechanisms of host-EHEC interactions [39]. Additionally, Heo and coworkers showed that 3D organoids derived from human small intestine represent physiologically relevant in vitro model system for *Cryptosporidium* infection [40]. In a more recent study, Roodsant and coworkers produced a human intestinal organoid-derived monolayer to elucidate intestinal host-pathogen interactions [41]. By optimizing the culture medium, Kozuka and coworkers made a functional monolayer from ileal organoids that was capable of hormone production and ion transport in response to external stimuli from various compounds [42]. These monolayer models show higher expression and function of major pharmacokinetic-related enzymes and transporters compared to the model derived from Caco-2 cells. The spherical organoids into 2D monolayers are not self-renewing. On the other hand, organoid-derived intestinal monolayers have *in vivo*-like structural and functional characteristics simultaneously. Polarity of the apical and basolateral region is maintained in organoid-derived intestinal monolayers. Thus, their profiles in the monolayers are close to those of human and mouse intestines in general. Accordingly, improved methods for generating intestinal organoid-derived monolayers will provide a general platform for a wide range of applications.

### 2.2. Co-Culture Model for 2D Epithelial Monolayers and ENS

The source of ENS cells within in vitro systems is obtained from dissociated cultured cells derived from adult ENS or immortalized cultured neurons [33,43]. As a great part of the ENS developmentally arises from the dorsal neural tube, a tissue-engineering approach with embryonic and induced pluripotent stem cells has been used to generate functional ENS [44,45].

In a more recent study, by using a novel, in vitro, transwell-based co-culture system, Puzan and coworkers showed that the ENS contributes to regulation of ISC fate [33]. In particular, chromogranin A-positive epithelial cells were increased when compared with or without ENS, suggesting enhancement toward the enteroendocrine cell differentiation [33]. The research provides a co-culture model for the study of enteric-epithelial crosstalk and is adaptable for specific investigation into ENS and intestinal health and diseases.

Remarkable progress has been made in understanding organoids using cell cultures such as embryonic stem cells (ES cells), induced pluripotent stem cells (iPS cells), adult stem cells including ISCs, and primary cultures cells derived from human tissues [46]. Previously, the Wells laboratory gained progress using the directed differentiation of human ES and iPS cells to make the composite structure and physiological 3D human intestinal organoids [47,48]. Recently, Workman and coworkers revealed that human ES and iPS cells were differentiated into neural crest and intestinal epithelial cells to generate ENS and epithelial sources independently for next co-culture system [45]. Of interest, after in vitro growth, the resulting ENS harbored morphology similar to the embryonic ENS, developmentally [48]. Furthermore, they confirmed that human iPS-derived neural crest cells formed complex ganglionic structures and interganglionic fibers that showed neuronal activity [45]. They also focused on understanding and targeting ENS components of intestinal disease such as Hirschsprung’s disease that is a developmental disorder characterized by the absence of ganglia in the distal colon. There is high enthusiasm for emerging stem-cell-based therapies for the treatment of enteric neuropathies [45,49]. The researchers developed a novel transwell-based model of the neuronal-epithelial microenvironment. However, whether IEC growth and differentiation in vitro are similar to their neuronal counterparts during in vivo growth remains unclear.

Tight regulation of IEC proliferation and differentiation is vital for intestinal homeostasis. Research linking neurotransmitters to this process is ongoing, and the mechanisms remain unclear. Many cholinergic neurons innervate the epithelium and release ACh to stimulate mucosal secretion [50]. Inhibition of ACh with simultaneous upregulation of serotonin (5-HT) signaling returns the crypt proliferation index to wild-type levels, suggesting that 5-HT effects on epithelial proliferation occur through a cholinergic neuronal intermediate [51]. Concerning ACh itself, increased ACh signaling in cultured intestinal organoids increases epithelial growth in an ENS-dependent manner, suggesting cholinergic control of epithelial dynamics [52]. Furthermore, the ablation of mAChR subtypes (M2, M3, and M5) increases small intestinal proliferation, suggesting an inhibitory influence of ACh on IEC proliferation [18]. Pharmacologically, ACh treatment of intestinal organoids decreases *cyclin D1* transcript expression, also corroborating an inhibitory role of ACh in IEC proliferation [53]. Cyclin D1 is a regulator of cell-cycle entry, and cyclin D1-null mice display reduced cell proliferation in intestinal crypts [54]. Especially since the M3 receptor gene is expressed in *Lgr5*^+^ stem cells, there is a need for further studies of ACh influence on stem cell behavior [55]. Collectively, ACh is an attractive molecule for IEC growth and differentiation.

## 3. ISC Function Controlled by Intestinal Epithelium

In the previous study, it has been suggested that ACh signaling controls intestinal physiological functions including intestinal epithelial ion transport in an ENS -independent manner [56,57]. After the binding of propionate to the apical short-chain fatty acid receptor (s), GFR41 (FFA3) and/or GPR43 (FFA2), on the colonocytes, ACh is basolaterally released from the colonocytes [58]. Then, the ACh secretion activates cystic fibrosis transmembrane conductance regulator (CFTR)-mediated chloride secretion through multiple signaling pathways [59]. As intestinal organoids have no nerve cells, we can explore the function of non-neuronal ACh itself. Therefore, intestinal organoids represent a good assay system for studying the role of non-neuronal ACh in cell proliferation and differentiation.

### 3.1. Intestinal Organoids for Research into the Non-Neuronal Cholinergic System

All components of the cholinergic system (for example, choline acetyltransferase (ChAT), ACh, mAChRs, nAChRs, choline transporter, vesicular ACh transporter (VAChT), and acetylcholinesterase) have been demonstrated in mammalian non-neuronal cells [60,61,62,63,64]. In our previous pharmacological studies using intestinal organoids, non-neuronal endogenous ACh released from intestinal epithelium was found to play a critical role in maintaining homeostasis and inhibiting the differentiation of *Lgr*5^+^ ISCs via mAChRs (M1, M2, and M3) [20]. How ISC proliferation, differentiation, and maintenance are controlled and which inductive signals are required for tissue maintenance are well established [65]. Obviously, ACh impacts IEC growth and differentiation via mAChRs including M3 in vivo, yet the exact mechanisms, sources, and conditions remain to be determined. Recently, we showed that the size and function of the intestinal stem niche increased is regulated by M3 signaling in a whole-body knockout of M3 (M3^−/−^) in mice [66]. In culture, intestinal organoids derived from M3^−/−^ crypts and single ISC could be rapidly-growing compared with wild-type (WT) organoids [66]. We then performed comparative gene expression profiling on WT and M3^−/−^ crypts using RNA sequencing, and found that erythropoietin-producing hepatocellular carcinoma cell receptor B (EphB) and its ligand (ephrin-B) composed of the EphB/ephrin-B signaling pathway are upregulated [66]. Pharmacologically, mitogen-activated protein kinase/extracellular signal-regulated kinase (MAPK/ERK) signaling inhibition by a MEK inhibitor (U0126) decreased excessive this signaling in M3^−/−^ crypts, leading to that of WT level [66]. Taken together, M3, EphB/ephrin-B, and the MAPK/ERK (MEK) signaling cascade get into alignment for maintenance of the homeostasis of IEC growth and differentiation, leading to modifications of the cholinergic intestinal niche (Figure 4A) [66]. In conclusion, the expression of M3 enables ISCs and transit-amplifying cells (progenitor cells) to fine-tune their cellular response upon ACh stimulation and ensure the maintenance of intestinal tissue homeostasis.

The identification of signaling pathways is thought to offer new insights into tissue homeostasis as well as oncogenesis. Most colorectal and small intestinal cancers overexpress M3, and in vivo studies have shown that genetic inhibition of M3 activity attenuates colon cancer growth and small intestinal adenoma formation in carcinogenic hypersensitive model mice (Apc^min/+^ mice) [67]. Though signaling proteins enhancing the proliferation of stem/progenitor cells are often encoded by proto-oncogenesis, EphB receptors by way of exception act as tumor suppressors by controlling cell migration and inhibiting invasive growth despite enhancing cell proliferation in the intact intestinal epithelium [68,69,70]. Analysis of Apc^min/+^ epithelium on the silencing of *EphB4* expression revealed an increase in cell proliferation, extracellular matrix remodeling, and EGF signaling [71]. In our study, high *EphB4* expression was observed in M3^−/−^ crypts [66]. Thus, M3 activation and its downstream signaling may be targets for potential therapies in colorectal cancer. Collectively, M3 and its downstream signal transduction pathway such as EphB/ephrin-B signaling may be novel therapeutic targets for overcoming the obstacles and accepting clinicians to exploit the nature of colorectal cancer cells.

As for the contribution of nAChR signaling to IEC growth and differentiation, it has been proven that ACh activates the α2β4 nAChR subtype localized with Paneth cells and modulates the expression of non-canonical Wnt ligands, Wnt5a and Wnt9b, using an intestinal organoid model (Figure 4B) [21]. This activates Wnt signaling through Frizzled receptors, resulting in enhanced proliferation and differentiation in the stem cell niche [21]. Recently, we demonstrated that ablation of *β4* subunit gene causes the crypt size decrease and subsequently decreases the number of ISCs and IECs, which is linked to upregulation of nAChR-driven Hippo and Notch signaling pathways [21,72]. Notch signaling (Notch1/2 receptors and delta-like ligand 1) is involved in the absorptive/secretory lineage switch [73]. The non-neuronal cholinergic system probably have a pivotal role in modulation of microenvironment outside of the ENS in the mouse intestine.

### 3.2. ISC Function Controlled by Tuft Cells

Tuft cells were first demonstrated in rat tracheal mucosal epithelium [74]. Although the cells were discovered more than sixty years ago, their exact functions have remained elusive. Tuft cells are thought to communicate with enteric neurons [75,76]. Stimulation of intestinal cultured organoids with co-culture with enteric neurons or by cholinergic drugs lead to an increase and survival of tuft cells in vitro [52]. On the other hand, the ablation of tuft cells leads to decrease organoid growth in vitro and prevents mice from repairing from chemically-induced injury in vivo [52]. These results indicate that there is a functional interaction between ENS and tuft cells. And, tuft cells play an important role for transducing signals between enteric nerves and epithelium. Interestingly, tuft cells express ChAT for the production of ACh in mice and humans (Figure 4B) [77,78]. However, it is unknown whether ACh storage via VAChT and synaptic vesicle release occurs within tuft cells [77,78].

Of interest, progenitors of tuft cells are found neighboring the ISC zone within the crypt, whose position is close to the position of *Bmi1*^+^ ISCs [79]. As localization of enteroendocrine tuft cells marked by doublecortin-like kinase 1 (DCLK1) are consistent with a transit-amplifying cell rich zone, it is suggestive of a specific function between tuft cells and quiescent *Bmi1*^+^ ISCs [80,81]. Recently, Middelhoff and coworkers found a novel interaction between M3 signaling and *Lgr5*^+^ ISC maintenance [81]. Their results indicate that DCLK1-positive (DCLK1^+^) tuft cells in the M3^−/−^ mice have an ability to compensate for the loss of M3 signaling by secreting ACh [81]. However, mature tuft cells located at the villi do not show any response to reduced cholinergic signaling [81]. Accumulating evidence indicates that non-neuronal ACh regulates *Lgt5*^+^ ISCs expansion via M3 [20,66,82]. Additionally, ablation of DCLK1^+^ cells inhibits the proliferation of epithelium [83]. Thus, DCLK1^+^ tuft cells have an important role in the regulation of IEC growth and differentiation in the ACh-M3 axis (Figure 4B). Another signaling pathway promoting cell proliferation and differentiation such as EGF signaling has been demonstrated to be regulated by muscarinic receptor (M1, M2, and M3) signaling [84,85,86,87]. Thus, tuft cells have the capacity to support *Lgr5*^+^ ISCs in part orchestrating intestinal cholinergic niche [81].

## 4. Application of Organoids in Regenerative Medicine

The recent development of intestinal organoid culture derived from healthy or affected tissue biopsies makes organoids a highly promising tool for medical research. Manipulation of organoids with many laboratory in vitro method are already contributing to basic science including in developmental biology and adult stem cell biology.

As a pioneer study, with the advent of organoid technology, Fukuda and coworkers showed that organoids derived from small intestinal crypts are able to reconstitute self-renewing epithelia in the colon [88]. In spite of common features, the epithelia of the small intestine and colon show many differences (Figure 1). In particular, Paneth cells reside only in the small intestine and play a role in innate immunity and the ISC niche [89]. Though the colon lacks Paneth cells, deep crypt secretory (DCS) cells are intermingled with *Lgr*5^+^ stem cells at the crypt bottom [90]. Recently, Sasaki and coworkers concluded that *regenerating islet-derived protein 4* (*Reg4*)-positive DCS cells serve as Paneth cell equivalents in the colon crypt niche (Figure 1) [91]. In a previous study, small intestinal epithelial progenitors of fetal origin were able to grow as fetal enterospheres in vitro and transplant into colonic epithelium [15]. Since then, the adult small intestinal epithelia have been found to be stably incorporated in the colon at four weeks post-transplantation [88]. Within some areas of grafts, there are structures reminiscent of the typical architecture of the small intestinal epithelium, and lysozyme-positive Paneth cells are found at the bottom of the crypts [88]. This is the first time that adult small intestinal epithelial cells in culture have been able to repopulate onto the colon in a manner different from that of fetal small intestinal progenitor cells. These findings highlight the epithelium-intrinsic program that allows for the maintenance of organ-specific stem cell properties in adult intestines.

Sugimoto and coworkers established an orthotopic xenograft system for the first time using organoid technology that permits indefinite expansion of the human large intestinal epithelium in regenerative medicine [92]. The group is also launching an organoid biobank of neuroendocrine neoplasms [93]. The application of organoid-based regenerative medicine such as the orthotopic transplantation may be modified for future clinical applications, especially to the treatment of colon cancer [94,95].

Moreover, their recent breakthrough using an organoid-based organ-repurposing approach has expanded culture studies and transplantation procedures [96]. By exploiting the structural similarities between the small intestinal and colonic subepithelia, they have used organoid transplantation to replace colonic epithelium with small intestinal epithelium to generate a “small intestinalized colon” [96]. Using human ileum organoids for xenotransplantation into immunodeficient mice [92], they have shown that the small intestinal organoids rebuild villus structures in the mouse colon [96]. To understand the spontaneous formation of villus-like structures from xenotransplanted human ileum organoids, they developed a unique 2D culture of small intestinal organoids with continuous flow [96]. Notably, this mechanical stimulation facilitated the formation of villus-like structures from a monolayer of duodenum and ileum organoids, but not of colon organoids [96]. Though exposure to the flow of culture medium has been found previously to induce the formation of villus-like structures in Caco-2 and in primary small intestinal epithelial cells [97,98], their results reveal that the epithelium of the small intestine has an inherent villus-forming program including a mechanical “flow” cue for its activation. During the course of regeneration, engrafted organoids initially formed crypt-like structure, and then developed villus architectures comparable with the histology of the mature small intestine [96]. This response is consistent with flow-dependent formation from the monolayer culture of human small intestinal organoids. This suggests that exposure to the bowel flow is able to drive the maturation of the small intestinal epithelium [99]. They conclude that their strategy represents an opportunity to bridge the gap between in vitro stem cell research and its translation to the treatment of lethal human diseases of the small intestine [96]. Other potential applications including the autologous and orthotopic engraftment of genetically corrected small intestinal organoids in case of genetic diseases such as cystic fibrosis or microvillus inclusion disease will be available. Undoubtedly, future studies will reveal the possible clinical applications of this approach.

## 5. Conclusions

The development of Matrigel-based 3D culture systems for the expansion of ISCs and their subsequent differentiation provides sufficient material for experimental testing. Traditional models of the intestinal epithelium relied on epithelial cancer cell lines, including Caco-2 and HT29 cells, which do not fully represent a healthy intestinal epithelium. These lines have a limited ability to represent the various epithelial cell phenotypes. Organoid-derived monolayer systems solve this problem.

Increasing numbers of studies suggest that epithelial-ENS communication influences intestinal homeostasis. The model developed by Puzan and coworkers provides a coculture system for the study of enteric-epithelial crosstalk, which is easily adaptable for specific perturbations and investigations of intestinal health and disease [33]. The initial characterization of this system demonstrates the impact of trophic cells on ISC differentiation and barrier integrity [33]. Though the crosstalk between the ENS and epithelium remains poorly understood, the field is poised to benefit from integration of neuroscience and epithelial techniques. The model will determine alterations in the functions of both epithelia and neurons due to pharmaceutical exposure, which affects both epithelial and neural function [100,101]. Furthermore, it would be interesting to explore if restoring functional epithelial-ENS interactions to aganglionic regions in Hirschsprung’s disease would result in normal intestinal physiology [102].

Heterotopic transplantation approaches using small intestinal stem cells in the colon provide evidence for the efficacy of cell therapy of intestinal diseases, such as short bowel syndrome [88,96]. Spontaneous formation of villus-like structures with continuous flow may further provide important discoveries in intestinal biology regarding epithelial-ENS interactions. Since small intestinal organoids have the capability for rapid growth and stable differentiation from a single biopsy-size epithelium sample, they may be a powerful tool for human disease research and precision therapy. Future studies will reveal whether the transplantation procedures can contribute to personalized medicine for individual patients.

## Figures and Tables

**Figure 1 ijms-22-10912-f001:**
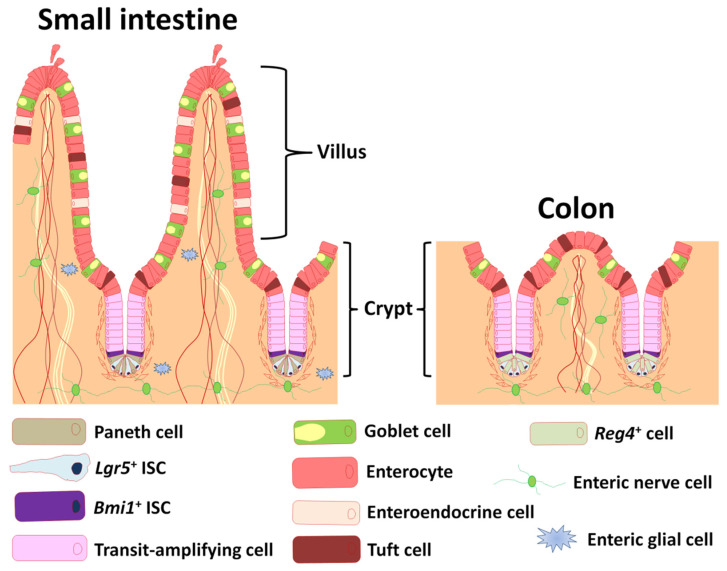
Depiction of the anatomy of intestinal epithelium and enteric nervous system. Cartoon depiction of small intestinal and colonic epithelium showing differentiated cell types, transit-amplifying cells, and enteric nerve and glial cells. Lgr5: leucine-rich repeat-containing G-protein coupled receptor 5, Bmi1: B-cell-specific Moloney murine leukemia virus integration site 1, Reg4: regenerating islet-derived protein 4.

**Figure 2 ijms-22-10912-f002:**
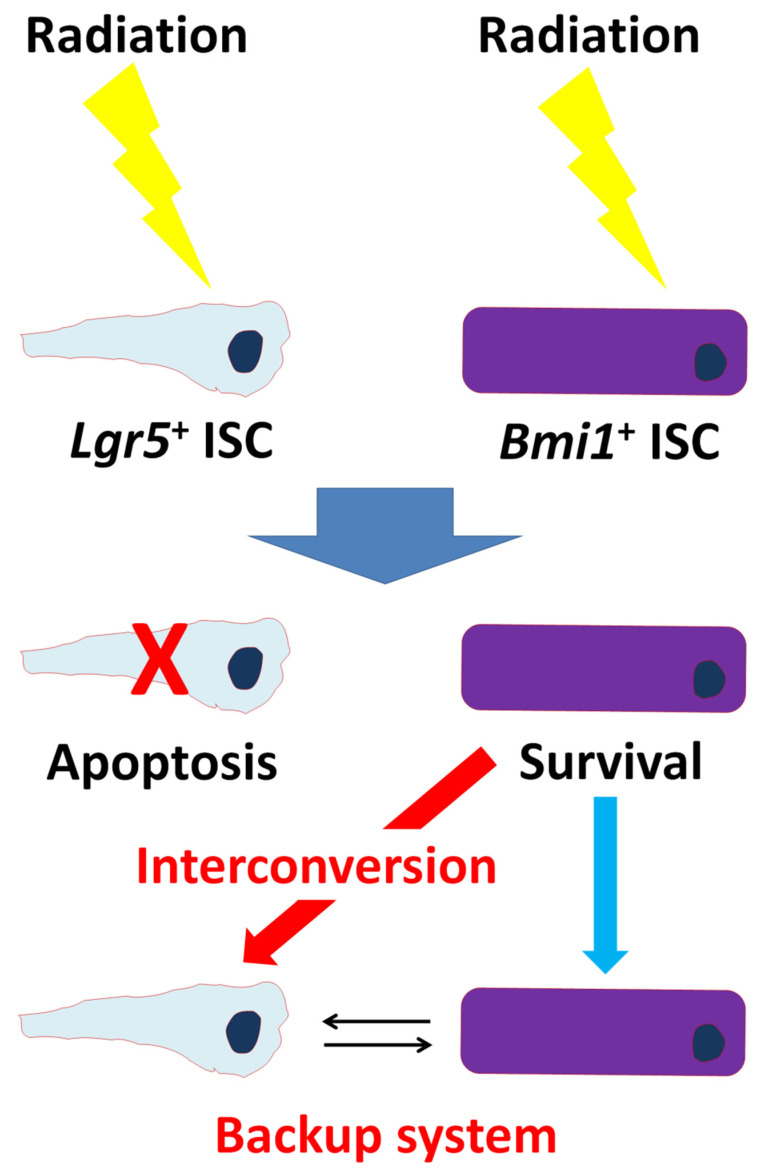
Backup system between *Lgr5*^+^ and *Bmi1*^+^ ISCs. Upon radiation, *Lgr5*^+^ ISCs undergo apoptosis, forcing *Bmi1*^+^ ISCs to assume a dominant role. During recovery from radiation, *Bmi1*^+^ ISCs give rise to *Lgr5*^+^ ISCs to restore homeostasis.

**Figure 3 ijms-22-10912-f003:**
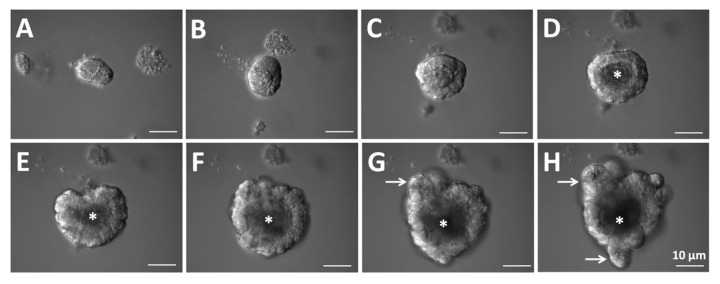
Representative images of a growing organoid derived from a crypt. (**A**–**C**) After the start of crypt culture, the upper opening rapidly become sealed. (**D**–**F**) Then, the lumen is filled with apoptotic cells. (**G**,**H**) The crypt region undergoes continuous budding events. (**D**–**H**) Asterisks indicate the lumen. (**G**,**H**) White arrows indicate crypt budding.

**Figure 4 ijms-22-10912-f004:**
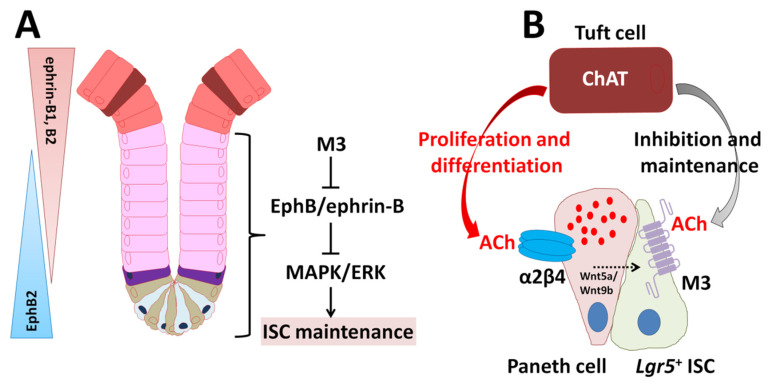
Model summarizing the importance of the cholinergic intestinal niche for the maintenance of IES homeostasis based on [65]. (**A**) Model depicting the proposed role of M3 through EphB/ephrin-B signaling in the ISC niche. (**B**) Non-neuronal ACh released from tuft cells activates the nicotinic receptor α2β4 localized in Paneth cells and the muscarinic receptor M3 localized in ISCs. α2β4 signaling modulates the expression and release of Wnt5a and Wnt9b. Eventually, proliferation and differentiation of ISCs are enhanced. The M3 signaling directly inhibits and maintains ISC proliferation and differentiation.

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
