# Peer review of "Modeling Intestinal Stem Cell Function with Organoids"

_ijms, 2021, doi:10.3390/ijms222010912_

Round 1

Reviewer 1 Report

The authors did a comprehensive review on the biology of intestinal stem cells in the intestinal organoids and the potential application of organoids. It covers all the important aspects of this field. The only concern comes from Figure 2, that the backup system between Lgr5+ and Bmi1+ ISCs is not very clear, i.g. survival and dead relationship indicates interconvert will kill the cells? Please redesign the figure.

Author Response

The authors did a comprehensive review on the biology of intestinal stem cells in the intestinal organoids and the potential application of organoids. It covers all the important aspects of this field. The only concern comes from Figure 2, that the backup system between Lgr5+ and Bmi1+ ISCs is not very clear, i.g. survival and dead relationship indicates interconvert will kill the cells? Please redesign the figure.

Answer: Thank you very much for the reviewer's comment concerning Figure 2. Figure 2 is complicated for readers. The damaged Lgr5+ cells are not killed by the survived Bmi1+ cells during interconversion. Though interconversion occurs between Lgr5+ and Bmi1+ ISCs, the precise mechanisms underlying the cellular plasticity remain to be determined. According to the reviewer's comment, we redesigned the figure.

Reviewer 2 Report

This is an interesting topic of study since organoids represent a step forward in in vitro experimentation.

However, I have a few concerns to consider this article suitable for IJMS:

  • 137 reviews on intestinal organoids in the last 5 years.
  • What is novel about this review?
  • What are the search criteria and the time period encompassed by this search?

Author Response

Reviewer 2

This is an interesting topic of study since organoids represent a step forward in in vitro experimentation.

However, I have a few concerns to consider this article suitable for IJMS:

  • Comment:137 reviews on intestinal organoids in the last 5 years.
  • What is novel about this review?
  • Answer: I understand that a large number of review articles concerning intestinal organoids are published in the last 5 years. It means that organoid technology is a highly promising tool for medical research as well as for basic science including in developmental biology and adult stem cell biology. The novel view about our review is that neuronal and non-neuronal ACh control ISC function including proliferation, differentiation, maintenance (i.g. Takahashi et al., 2021; Middelhoff et al., 2020; Puzan et al., 2018) and is that organoid technology is useful for an organoid-based organ replacement approach for intestinal diseases (i.g. Sugimoto et al., 2021).

  • What are the search criteria and the time period encompassed by this search?
  • Answer: The search criteria in our review article are as follows. And, the time period encompassed by our search is from 2014 to 2021 in the last 7 years.

(1) We summarize recent findings of ISC research with special emphasis on use of intestinal organoids.

(2) We focus on the cholinergic system involved in enteric nerve, intestinal epithelial cells, and intestinal stem cells, which are attracting a great deal of attention.

(3) We also review the potential application of intestinal organoids for regenerative medicine such as recent organoid-based organ-repurposing approach.

Reviewer 3 Report

In this manuscript by Takahashi et al, the authors summarized recent findings of intestinal stem cell (ISC) research, with special emphasis on uses of organoids. This is a well-written review with an organized flow and useful up-to-date information. However, there are a number of concerns that need to be addressed before this manuscript is in a publishable fashion. Specific comments are as follows:

1. There are obvious beneficial aspects of using organoids over monolayer cells as described in other literature. The authors reviewed a number of studies using organoid-derived monolayers to resolve the problem that test materials cannot reach the lumen of organoids. How are these monolayered cells different from monolayers formed from isolated ISCs? Additionally, do these cells grown in 2D have different culture conditions from 3D organoids. Do they retain similar behavior such as polarity?

2. The authors used "ISC function" in the section titles. We know that ISCs replenish of differentiated intestinal epithelial cells in physiological conditions. How about in experimental conditions? The authors briefly described proliferation and differentiation. What are the measurements that represent ISC function? How differentiated can the ISCs be in experimental conditions?

3. A brief introduction about function of different IEC cell types is suggested.

4. Are IESs and IECs the same thing? Both terms are used in different parts of the article.

Author Response

Reviewer 3

In this manuscript by Takahashi et al, the authors summarized recent findings of intestinal stem cell (ISC) research, with special emphasis on uses of organoids. This is a well-written review with an organized flow and useful up-to-date information. However, there are a number of concerns that need to be addressed before this manuscript is in a publishable fashion. Specific comments are as follows:

  1. There are obvious beneficial aspects of using organoids over monolayer cells as described in other literature. The authors reviewed a number of studies using organoid-derived monolayers to resolve the problem that test materials cannot reach the lumen of organoids.

Comment: How are these monolayered cells different from monolayers formed from isolated ISCs?

Answer: The spherical organoids into 2D monolayers are not self-renewing. It suggests that stem cells are lost over time. On the other hand, organoid-derived intestinal monolayers have in vivo-like structural and functional characteristics simultaneously.

Comment: Additionally, do these cells grown in 2D have different culture conditions from 3D organoids.

Answer: Yes. These cells grown in 2D have different culture conditions from 3D organoids. To boost intestinal epithelial proliferation, the culture medium with intestinal subepithelial myofibroblasts-conditional medium (ISEMF_CM) and Wnt3a are used.

Comment: Do they retain similar behavior such as polarity?

Answer: Yes. Polarity of the apical and basolateral region is maintained in organoid-derived intestinal monolayers. Upon basolateral administration of ISEMF_CM and Wnt3a, confluent epithelial monolayers with in vivo-like cell composition and distribution are reproducibly obtained on Transwells inserts. As the measured TEER values are within the expected physiological range, an adequate maturation of the epithelium is obtained.

  1. The authors used "ISC function" in the section titles. We know that ISCs replenish of differentiated intestinal epithelial cells in physiological conditions.

Comment: How about in experimental conditions? The authors briefly described proliferation and differentiation.

Answer: We will explain 3D organoid culture condition as follows. Isolated crypts are mixed with Matrigel and plated in 24-well plates. After Matrigel polymerization, 500 μl crypt culture medium (Advanced Dulbecco’s Modified Eagle’s Medium/F12) containing growth factors (20 ng/ml mouse EGF), 500 ng/ml R-spondin 1, and 100 ng/ml Noggin is added. The organoids are maintained at 37°C in a humidified atmosphere with 5% carbon dioxide. Culture medium is changed every other day.

Comment: What are the measurements that represent ISC function?

Answer: Usually, we use ISC-specific marker genes, Lgr5, Ascl2, and Olfm4 in ISC proliferation. On the other hand, we use epithelial cell markers for Paneth cells (lysozyme), enterocytes (villin), goblet cells (mucin-2), and enteroendocrine cells (chromogranin A) in ISC differentiation.

Comment: How differentiated can the ISCs be in experimental conditions?

Answer: To boost ISC differentiation in experimental conditions, 500 μl crypt culture medium (Advanced Dulbecco’s Modified Eagle’s Medium/F12) containing growth factors (20 ng/ml mouse EGF), 500 ng/ml R-spondin 1, and 100 ng/ml Noggin is added. Additionally, Paneth cells are the source of multiple stem cell growth factors (for example, Wnt3, EGF, and TGF-α) which are essential signals for stem cell maintenance (Sato et al, 2011). Indeed, co-culturing of sorted ISCs with Paneth cells dramatically improves organoid formation (Sato et al, 2011).

Comment: 3. A brief introduction about function of different IEC cell types is suggested.

Answer: Thank you very much for your suggestion. I added a brief introduction about function of different IEC types as follows. 1) “absorptive" enterocytes. 2) “mucus secreting" goblet cells. 3) “hormone secreting" enteroendocrine cells. 4) “Immune and neuronal" tuft cells. 5) “Wingless/Int (Wnt) secreting" Paneth cells.

Comment: 4. Are IESs and IECs the same thing? Both terms are used in different parts of the article.

Answer: Thank you very much for your comment. IECs is a correct abbreviation, not IESs. We corrected them in our revised review manuscript.

Round 2

Reviewer 2 Report

I still do not see clearly the novelty of this revision since the first part (up to page 5 of 13) does not provide any novelty on the formation of the organoids and the structure of the small intestine. This fact is reflected in the fact that the first 27 references used in the text are more than 5 years old.

It seems to me to be a review that is by all means scarce and that goes far beyond the objectives that have been previously stated both in the abstract and in the introduction:

1) Neuronal innervation and non-neuronal control of small intestinal crypts and its impact on the proliferation of ISCs in their proliferation, differentiation and maintenance. 
2)Organoid-based organ replacement for intestinal diseases.

Author Response

Comment: I still do not see clearly the novelty of this revision since the first part (up to page 5 of 13) does not provide any novelty on the formation of the organoids and the structure of the small intestine. This fact is reflected in the fact that the first 27 references used in the text are more than 5 years old.

Answer: In 2009, Sato and coworkers first revealed intestinal organoid technology. The technology was an epochal event like iPS cell established by Yamanaka and coworkers in 2007. As mentioned in the revised manuscript, organoid technology has already been reviewed in detail elsewhere. To understand intestinal organoids and their original tissue (structure of small and large intestine) for wide range of readers, we need to introduce them at the first part (up to page 5 of 13) using the first 27 references in the revised manuscript. As my wish, I would increase the number of researchers focusing on organoid technolody and make effort to contribute it through the revised manuscript.

It seems to me to be a review that is by all means scarce and that goes far beyond the objectives that have been previously stated both in the abstract and in the introduction:

Comment: 1) Neuronal innervation and non-neuronal control of small intestinal crypts and its impact on the proliferation of ISCs in their proliferation, differentiation and maintenance.

Answer: The IECs dose not function by themself, but interacts with the ENS. Our review is intended to serve as a reference for both neuronal and epithelial biologists interested in studying their interactions. We show that ENS influences on epithelial proliferation and differentiation within stem and differentiated cells using multidisciplinary approaches (for example: 3D organoid-derived 2D monolayer technology) to uncover their interaction in the revised manuscript. We focus on non-neuronal control of small intestinal crypts by acetylcholine (ACh). It has long been hypothesized that ACh mediates the epithelial response through muscarinic receptors in the mouse. In addition, ACh signaling through activation of muscarinic ACh receptors regulates expression of specific genes that mediate and sustain proliferation, differentiation, and homeostasis in the intestinal crypts. Although mounting evidence indicates the pivotal role of ACh signaling in the regulation of intestinal cell homeostasis, the underlying mechanism remains elusive. Our and Middelhoff group newly report that endogenous ACh affects the size of the intestinal stem niche via M3 signaling (Middelhoff et al. 2020; Takahashi et al. 2021). We think that the research will influence on future study concerning ISC biology.

Comment: 2)Organoid-based organ replacement for intestinal diseases.

Answer: We think that organoid-based epithelial replacement approaches in intestinal diseases, for example, short bowel syndrome (Sugimoto et al. 2021) is fairly comprehensive and up-to-date. We describe the discussion of heterotopic transplantation of small intestinal organoids into the colon, or of orthotopic xenografts in human colon. To further deepen the discussion in our revised manuscript concerning organoid-based organ replacement for intestinal diseases, we added new sentences as folows.

(New sentences)

Other potential applications including the autologous and orthotopic engraftment of genetically corrected small intestinal organoids in case of genetic diseases such as cystic fibrosis or microvillus inclusion disease will also be available.